# Research on low-Impact development layouts in old urban residential communities based on multi-objective optimization

Fuqin Liu, Yijun Fan, Xinran Liu, Yanjuan Li *

School of Civil Engineering, Architecture, and Environment, Hubei University of Technology, Wuhan, Hubei, China

* liyanjuan@hbut.edu.cn

## Abstract

Challenges such as poor drainage systems and limited space for renovation often arise during the sponge city transformation of old urban residential communities. This research focuses on an old residential community in Beijing utilizing the SWMM in combination with the NSGA-II algorithm. A Pareto optimal curve was identified iteratively, with construction costs, runoff reduction effects, and node overload time as the optimization objectives and the maximum area for the low impact development (LID) layout as a constraint. After transformation using LID measures, the runoff reduction rate is directly proportional to the construction cost. However, the rate of runoff reduction increases marginally when the construction cost exceeds 1.4 million yuan. Additionally, the average node overload time decreases as the construction cost and runoff reduction rate increase. Therefore, selecting a transformation scheme requires comprehensive consideration of factors such as budget and land use type in practice.

## Introduction

Rainstorms, characterized by their brief duration and spatial dispersion, lead to urban waterlogging due to the interplay of various factors, including precipitation, drainage infrastructure, and hydrodynamic conditions, which vary significantly across regions [1]. Rapid urbanization in China has led to extensive paving and modification of land surfaces in residential areas to satisfy living requirements, altering natural surface conditions. This has impacted the flow of surface and subsurface water and the evaporation of rainwater in urban environments. Studies have shown that alterations in surface conditions not only increase the vulnerability of cities to rainstorms but also contribute to their frequency. A decrease in vegetation cover shortens the evaporation time of rainwater, resulting in suboptimal evaporation outcomes. Furthermore, the proliferation of impermeable surfaces reduces rainwater infiltration, leading to lower groundwater levels and increased surface runoff [2].

**Data availability statement:** You can access the extracted data through the following Figshare link: https://doi.org/10.6084/m9.figshare.28188725.v1.

**Funding:** This research was funded by National Natural Science Foundation of China (grant number 42101375) and The APC was funded by Fuqin Liu,and the funder(National Natural Science Foundation of China) had no role in study design, data collection and analysis, decision to publish, or preparation of the manuscript, Fuqin Liu had role in decision to publish.

**Competing interests:** The authors have declared that no competing interests exist.

To effectively reduce surface runoff and control runoff pollutant loads, China has proposed the construction of sponge cities based on the concept of low impact development (LID) to achieve the goal of urban stormwater management [3,4]. In sponge city initiatives, LID facilities such as rainwater gardens, bioretention ponds, and permeable pavements are commonly used. To achieve the goals of runoff control and pollutant reduction, it is typically necessary to combine different LID facilities at the block or district level. However, due to the large number of sub-catchment areas, factors such as the number, area, and location of LID facilities may affect spatial layout plans, thereby influencing the accessibility and cost of flood mitigation, water quality improvement, and other construction goals in sponge cities [5,6].

Developed nations are at the forefront of urbanization, leading them to pioneer stormwater management and establish sophisticated stormwater management systems. The Storm Water Management Model (SWMM), introduced by the U.S. Environmental Protection Agency in 1971, remains one of the most extensively used and accurately developed comprehensive urban stormwater models [7]. Through continuous refinement and development, the SWMM has been instrumental in simulating rainfall runoff and sewage processing, forecasting flood-prone urban areas, and issuing early warnings. Additionally, it plays a crucial role in evaluating pollution contributions from various urban sources, both point and nonpoint, serving as a vital reference for urban drainage planning, waterlogging prevention, and pipeline system optimization [8]. Researchers worldwide have utilized the SWMM to incorporate LID facilities and their combinations, studying their impact on runoff and pollution reduction. By integrating LID controls into the SWMM, the model automatically manages the runoff distribution to LID facilities and calculates its effectiveness. The influence of LID controls on sub-catchment runoff was assessed by comparing simulation outcomes before and after their implementation. Randall et al. [9] found through simulation that combined LID application performed significantly better than single application, and under a certain transformation ratio, the combined LID layout could increase the total runoff control rate from 59.9% to 82.2%. Kim and Kim [10] used SWMM to establish a rainwater flood model with various LID measures, finding that the model could effectively reduce runoff and peak flow and that bioretention ponds exhibited better effects than permeable pavements and green roofs. Fan et al. [11]adopted the SWMM to construct a drainage model for a central urban area in China and compared the reductions in surface runoff, overflow nodes, and overloaded pipelines in various LID combination schemes through simulation experiments. In previous studies, the areas of different LID facilities were often set manually, and simulations were performed on various combinations of LID facilities to obtain comparative schemes for assessing the effects of different LID measures on reducing rainwater [12]. However, this method is influenced by subjective experience to a large extent, and the combinations of these methods are limited. Hence, this method cannot provide scientifically rigorous layout references for the studied community.

With the development of computer science, the merging of algorithms with models has addressed the complex application challenges of sponge cities. Rathnayake [13] considering the migration behavior of storms in control of urban sewer networks and

incorporating on-line storage tanks into the SWMM, then calling the NSGA-II algorithm to optimize, analysis shows that the developed multi-objective optimization control algorithm has been improved under the action of storage tanks. Dang [14] combined SWMM with Non-dominated Sorting Genetic Algorithm II (NSGA-II) to establish a multi-objective optimization model with TSS reduction and scheme construction cost as the control objectives and studied the optimal combination layout scheme of green roofs, permeable pavements, and rainwater buckets. Kumar et al. [15] combined the SWMM and NSGA-II models to establish a multi-objective optimization decision-making framework to examine the performance of best management practices (BMPs)/LID. Taking two cities in central Delhi, India, as examples, they analyzed the cost-effectiveness of LID facilities within the watershed and found that the optimal solution could effectively reduce urban runoff while maintaining the implementation cost and scale of BMPs/LID. Sun et al. [16] from China established an optimization model of the NSGA-II based on fast classification, taking the construction cost, comprehensive rainwater runoff coefficient, and comprehensive pollutant control rate as the objective functions. They leveraged the construction scale of source emission reduction facilities and sponge city runoff control indicators as the boundary conditions to formulate the optimal design scheme. Simulating and transforming a single target is far from sufficient, so most current studies focus on considering multiple optimization objectives simultaneously and selecting the most suitable set of construction plans for the optimized design of the study area.

The integration of the NSGA-II with the SWMM enhances the calibration of model parameters, leading to improvements in the model's precision and accuracy. Furthermore, this combination accounts for the uncertainty and randomness of parameters, yielding more robust optimization results and ensuring the reliability of the research findings. In this study, rainfall flood management was combined with the theory of multi-objective optimization to establish a multi-objective optimization and decision-making model for LID scale schemes. Source files from SWMM were extracted and compiled using programming languages to identify a Pareto optimal solution that balances construction costs, runoff reduction effects, and node overload times for LID facilities within a community, subject to specific constraints. By simulating various rainfall scenarios, feasible solutions were identified and selected as necessary to transition the residential community toward a sponge community. The approach of combining LID practices, as demonstrated in this work, serves as a valuable reference for communities with similar geographic conditions in northern China, facilitating the adoption of effective stormwater management strategies.

## Overview of the research area and methodology

### Overview of the research area

This study focused on a residential community situated in the Malianwa region, northwest of Haidian District, Beijing, within the Qinghe River Basin. This area experiences a continental monsoon climate characterized by distinct seasons throughout the year. The annual average rainfall totals 628.9 mm, with the majority occurring during the summer months (June to August), which receive 465.1 mm of precipitation, representing 70% of the annual total. The lowest amount of precipitation was recorded in the winter months (December to February). The community's terrain is generally flat, with slight variations in elevation ranging from approximately 47.23 m to 47.62 m. The total management area of the community spans 24,800 m$^2$, comprising 5,500 m$^2$ of rooftops, 4,700 m$^2$ of roads (with 3,800 m$^2$ of these roads being paved), and 14,600 m$^2$ of green spaces.

### Research methodology

To address the challenge of optimizing multiple objectives simultaneously, these objectives can either be consolidated into a single-objective optimization problem through weighting, or one specific optimization objective can be directly managed by transforming it into a constraint. Nonetheless, the global maximum or minimum solutions derived from this approach are significantly influenced by human understanding and cognition, which can lead to less than optimal optimization

 

performance [17]. Consequently, intelligent algorithms have been adopted for optimization purposes. NSGA-II, an evolution of traditional genetic algorithms (GAs), enables the simultaneous optimization of multiple conflicting objectives, thus improving the efficiency and quality of decision-making. This algorithm has found applications in the placement of logistics warehouses and shelters [18–20], as well as in determining optimal routes [20]. In the field of flood research, optimization goals may include reducing flood inundation areas, minimizing pollutant emissions, and maximizing rainwater collection and utilization. In the present study, an optimal combination solution for the layout of LID facilities, specifically a Pareto solution, was identified through continuous screening and the preservation of elite populations. This was achieved by implementing rapid non-dominated sorting, setting probabilities for crossover and mutation, and determining the total number of iterations. The algorithm's effective, non-dominated sorting mechanism, rank-based selection, and evolutionary strategies render it adept at handling complex optimization scenarios. This approach effectively mitigates the issue of local optima to a certain extent [21], as illustrated in Fig 1.

### Research content

**Generalization of the pipe network.** The research area includes a total area of 24,800 m². Based on the land use function, the area under study is segmented into residential, office, outdoor activity, and green spaces. The actual configuration of the rainwater drainage system was sourced from the rainwater collection layout plan. To enhance the clarity and intuitiveness of the final research model, the representation of the pipe network and the data of the study area were generalized. This process involved simplifying the shape and quantity of pipelines, retaining only those pipes that have a direct impact on drainage outcomes [22]. Consequently, the simplified model comprises a total of 58 rainwater pipes, 57 rainwater wells, and 2 discharge outlets, as depicted in Fig 2.

The main rainwater and sewage trunk pipes run through the residential area of the community from north to south, with branch pipes installed in alignment with the spacing between buildings. Each residential building is linked to the main rainwater trunk pipe through these branch pipes. The drainage pipes have circular cross-sections, with diameters

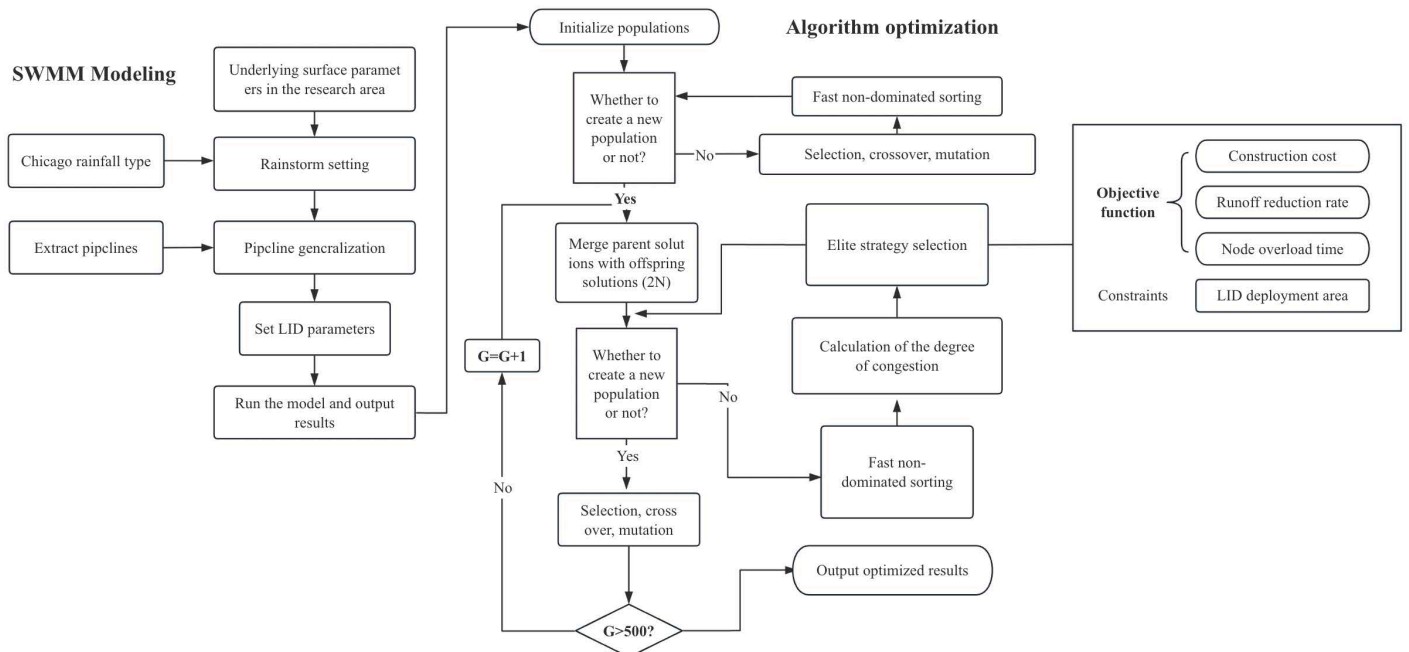

**Fig 1. Combination of SWMM software and the NSGA-II algorithm.**

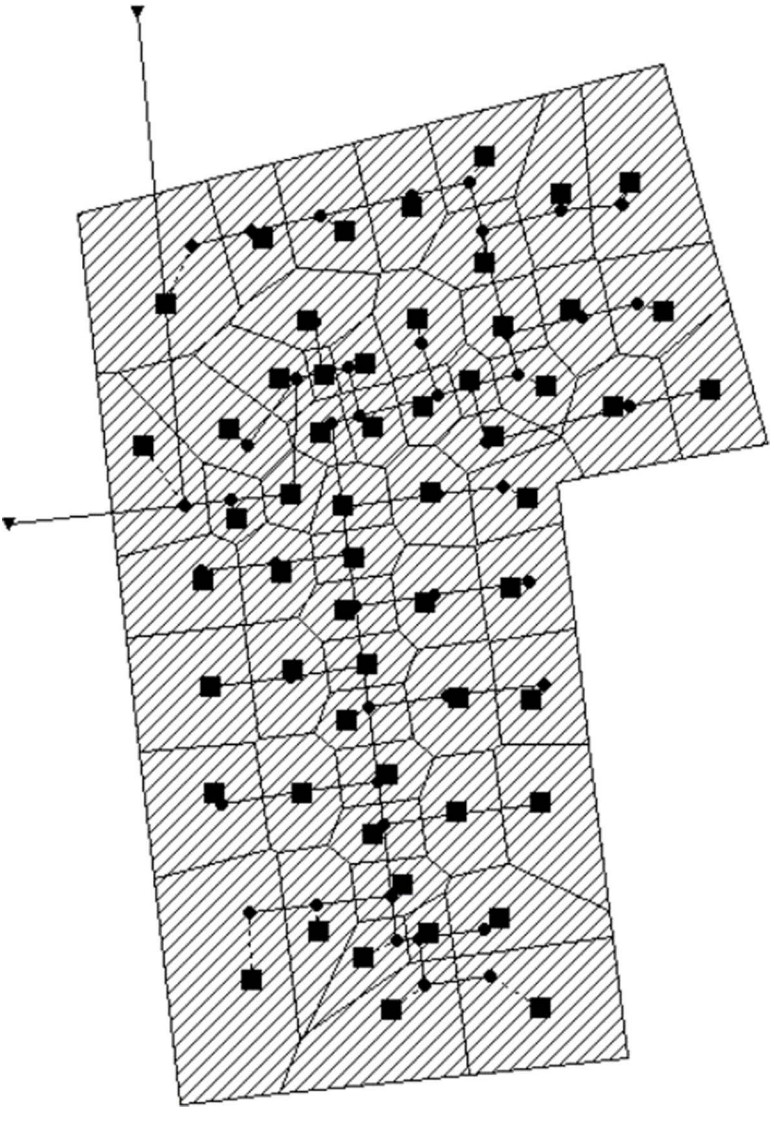

**Fig 2. Generalized Pipe Network.**

progressively decreasing from the main trunk pipe to the branch pipes. The slope of the pipes is determined according to the provided drawing specifications. The depths of the rainwater wells are set based on the elevation levels at the bottom ends of the drainage pipes, with the profiles of these pipes illustrated in Fig 3.

The system's discharge outlets serve as the terminal nodes of the rainwater drainage network. The model's simulation continues until the rainwater reaches these discharge outlets. In the context of this case study, the two discharge outlets are municipal rainwater wells situated to the north and west of the residential area.

**SWMM modeling.** The generalized rainwater model was imported into the SWMM, and the community was segmented into 57 catchment areas using the Thiessen polygon method to facilitate a more scientific analysis of drainage within the research area. Each catchment area functioned as a nonlinear reservoir, was associated with a specific rainwater well, and possessed unique rainfall-runoff and hydrological characteristics. According to the calculation protocols defined in the SWMM software, runoff from pervious and impervious surfaces was directed into rainwater wells, then flowed into branch and trunk pipes, and finally discharged into the outlet.

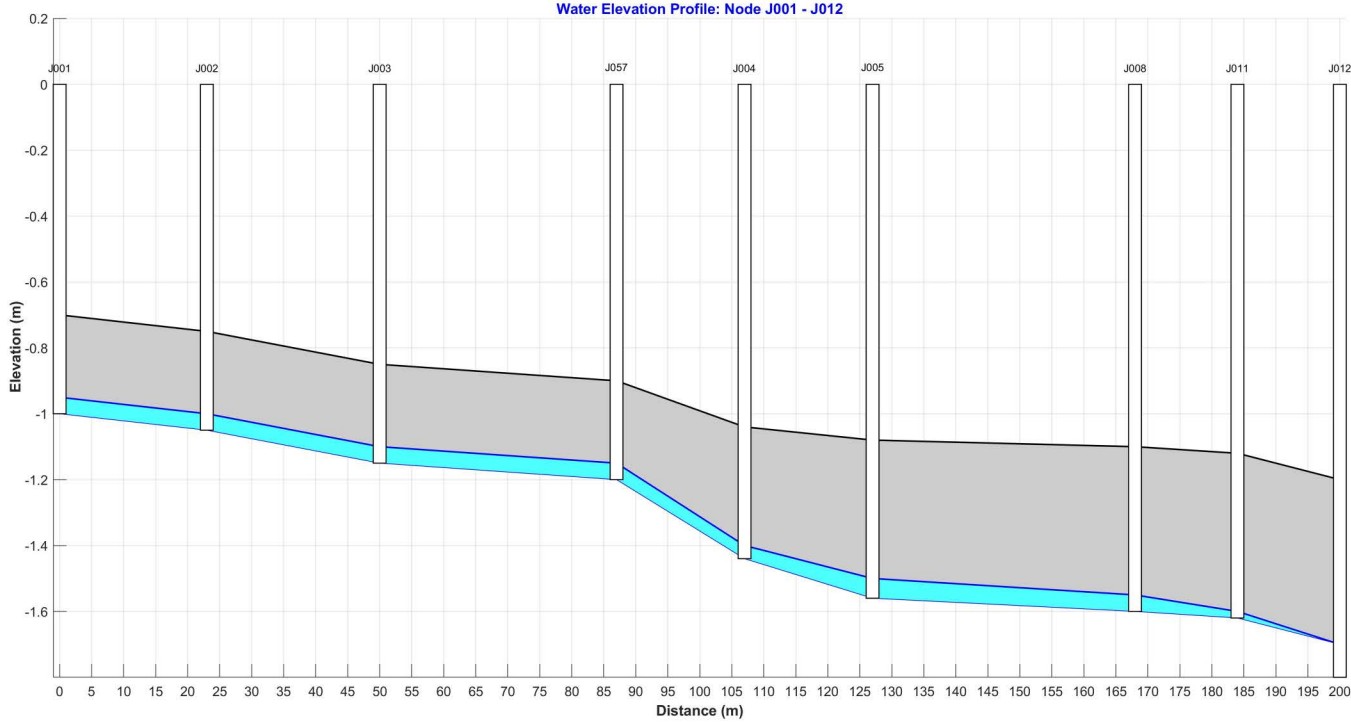

**Fig 3. J001-J012 Pipeline Section Diagram of Node.**

Parameters related to the underlying surface in each catchment area (Table 1) were established based on guidelines from the SWMM user manual and studies relevant to the surface conditions in Beijing [23].

The ratio of pervious to impervious areas within each catchment area was established based on the actual layout of the buildings depicted in the drawings. Given the flat terrain of the community, which lacks significant undulations, the slopes within the catchment areas are gentle. For simulating rainfall infiltration, the Horton equation (1) was employed for calculations, wherein the infiltration rate decreases gradually to a minimum value as rainfall persists. The model parameters were defined in terms of the maximum and minimum infiltration rates. Model calculations were performed under ideal conditions, assuming no initial accumulation or groundwater flow.

$$F(t) = f_c + (f_0 - f_c) \cdot e^{-\beta t},$$ (1)

$F(t)$—Infiltration rate;
$f_c$—Steady-state infiltration rate
$f_0$—Initial infiltration rate;
$t$—Time;
$\beta$—Empirical constant related to time and soil properties.

The characteristic width of a catchment area is defined as the ratio of the catchment area's surface area to the length of the longest overland flow path within that area. However, it is difficult to accurately determine the length of overland flow due to extensive vegetation coverage, diverse vegetation types, and undulating ground in practical engineering. For convenience in calculations, the SWMM automatically adopted the average length of multiple overland flow paths.

**Table 1. Underlying Surface Parameters in the Research Area.**

| Parameter | Meaning | Value |
|---|---|---|
| N-Imperv | Manning coefficient for impervious areas, s/m$^{1/3}$ | 0.013 |
| N-Perv | Manning coefficient for pervious areas, s/m$^{1/3}$ | 0.2 |
| Dstore-Imperv | Depression storage for impervious areas, mm | 2.5 |
| Dstore-Perv | Depression storage for pervious areas, mm | 5 |
| Infiltration mode | Infiltration mode | HORTON |
| MaxRate | Maximum infiltration rate, mm·h$^{-1}$ | 82 |
| MinRate | Minimum infiltration rate, mm·h$^{-1}$ | 3.3 |
| Decay Constant | Decay constant | 4 |
| Subarea Routing | Subarea routing method | OUTLET |

## Rainstorm setting

The Chicago rainfall type, known for its short duration, was utilized to simulate rainfall events. Rainstorms of short duration are notably intense and exhibit distinct characteristics across various climate [24]. Consequently, Beijing and its surrounding counties were divided into two rainstorm zones based on the local standards of Beijing [25], with township-level administrative divisions serving as the criteria for this classification. The community under study is situated in Zone II. The formula for rainstorm intensity specific to Zone II, as outlined in the local standards, was adapted for application in this research.

$$q = \frac{1602 \times (1 + 1.037 \times \log P)}{(t+11.593)^{0.681}},$$

(2)

q—Rainstorm intensity, L/(s*ha);
P—recurrence period, years;
t—Rainstorm duration, min;
As required for this research, rainfall data with recurrence periods of 1 year, 3 years, 5 years, and 10 years were used for testing. The precipitation histograms for various recurrence periods are depicted in Fig 4.

## LID Selection and parameter setting

LID measures involve employing a variety of green vegetation, biological elements, and innovative permeable concrete materials to increase rainwater infiltration and storage capabilities, reduce runoff discharge, and decrease the risk of urban flooding. LID construction in sponge cities is an urban planning and construction method based on the concept of environmental protection. By imitating the natural water cycle and adopting a series of sustainable technologies and environmental protection materials, rainwater is guided to flow toward vegetation, which slows surface runoff, reduces flood risk, and maintains ecological balance. Starting from the current situation of the residential community under study, four individual measures with minimal impacts on the ecological environment, namely, green roofs, permeable pavements, rainwater tanks, and sunken green spaces, are selected for renovation.

Common LID facilities can be classified into five types based on their functionality: infiltration, storage, regulation, conveyance, and purification. When selecting LID strategies, it is crucial to assess the specific conditions of the site and the functional characteristics of each type of LID facility, opting for either a single or a combination of measures. For example, in smaller sub-basins, large-scale facilities such as rainwater gardens or ground infiltration systems may not be suitable; in areas experiencing frequent rainfall, the implementation of additional rainwater gardens or rain barrels might be necessary

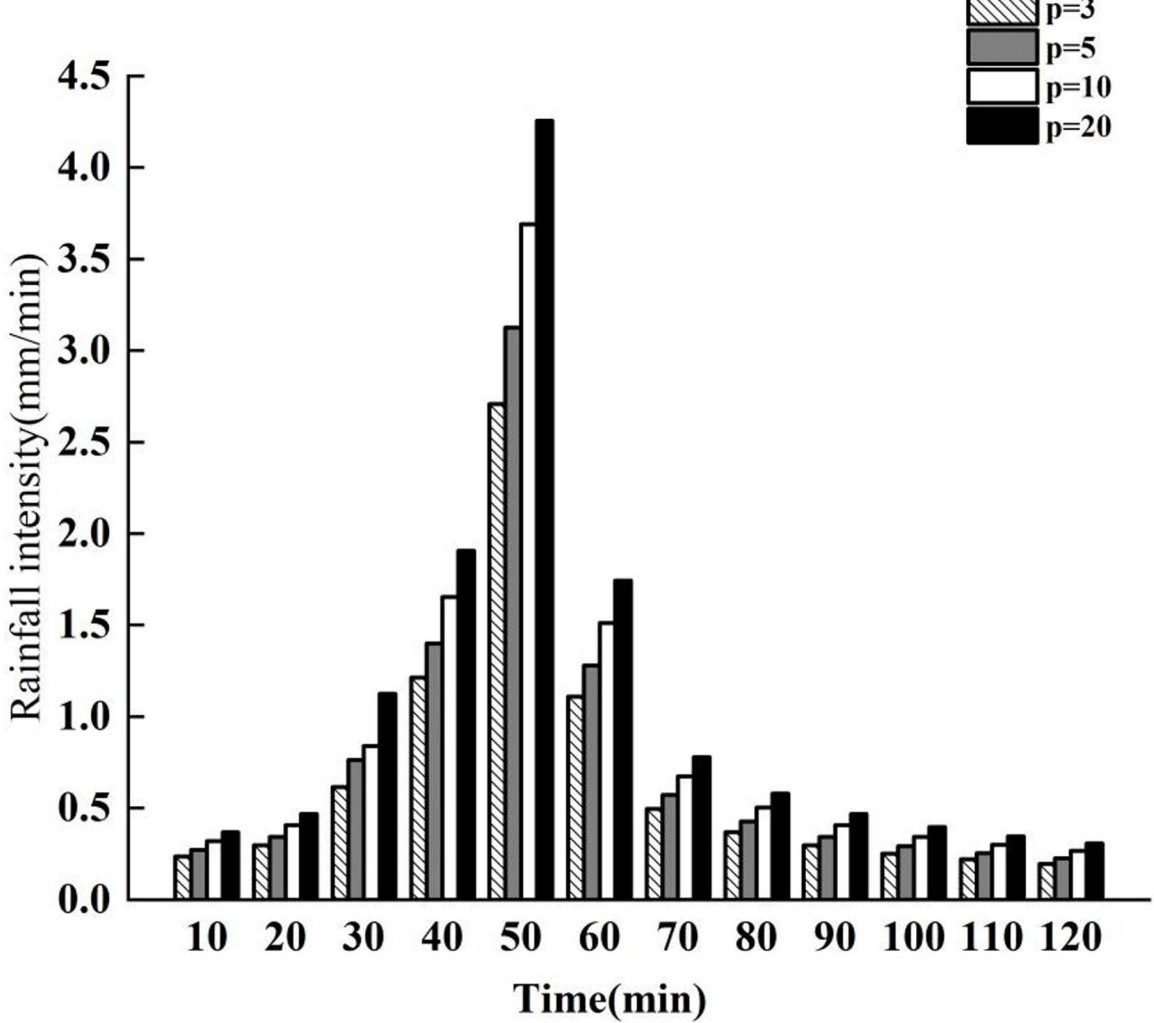

**Fig 4. Bar chart of precipitation during different recurrence periods.**

to manage water effectively [26]. Given that the research area is an old residential district, considerations for maintaining the living environment and dense land use are paramount during the planning and design stages.

In this study, the selected LID facilities included green roofs, sunken green spaces, rainwater tanks, and permeable pavements. Green roofs are designed to add vegetation to rooftops, thereby reducing rainwater runoff rates. This solution, however, requires careful consideration of roof slopes, waterproofing, and spatial conditions. In older residential areas, green roofs are widely feasible due to the extensive roof areas and residents' environmental awareness. As shown in Fig 4, common types of permeable pavements consist of permeable bricks and permeable concrete, which are not only easy to install but also suitable for extensive application. These pavements are typically used in areas that bear light loads, such as plazas, sidewalks, and paths for nonmotorized vehicles, and must be installed following regulations. The use of sunken green space is a strategy that transforms existing green areas by adding soil and vegetation to enable rainwater infiltration into the ground. This method is broadly applicable and cost-effective but is not ideal for regions with significant surface elevation differences. The rainwater tank, a storage-type LID facility, efficiently collects rainwater within a

**Table 2. Parameters of LID Facilities.**

| Layer | Facility, unit | Green Roof | Permeable Pavement | Rainwater tank | Sunken green space |
|---|---|---|---|---|---|
| **Surface layer** | Water storage depth | 50 | 20 | 5000 | 250 |
| | Vegetation coverage rate | 0.8 | / | / | 0.8 |
| | Surface roughness | 0.24 | 0.015 | | 0.03 |
| | Surface slope | 1 | 1 | | 0.2 |
| **Drainage layer** | Thickness of the pervious layer | 200 | / | / | / |
| | Porosity | 0.5 | | | |
| | Roughness | 0.1 | | | |
| **Water storage layer** | Thickness of the pervious layer | / | 450 | / | / |
| | Porosity | | 0.63 | | |
| | Clogging factor | | 8 | | |
| | Seepage rate | | 0 | | |
| **Soil layer** | Thickness of the pervious layer | 150 | 150 | / | / |
| | Porosity | 0.3 | 0.5 | | |
| | Water content in the field | 0.2 | 0.2 | | |
| | Wilting point | 0.1 | 0 | | |
| | Hydraulic conductivity | 8 | 0.5 | | |
| | Hydraulic slope | 10 | 10 | | |
| | Water head | 110 | 3.5 | | |
| **Pavement layer** | Thickness of the pavement layer | / | 100 | / | / |
| | Porosity | | 0.16 | | |
| | Permeability | | 254 | | |

**Table 3. Ratio Settings of LID Facilities.**

| Area | Classification | Area, 10,000 m² | LID facility | Maximum LID Ratio (of total) | Area covered, 10,000m² |
|---|---|---|---|---|---|
| **Roof** | Residential building roof | 0.48 | Green roof | 60% | 0.41 |
| | Office building roof | 0.07 | Green roof | 15% | |
| **Road** | Paved roads | 0.38 | Permeable pavement | 70% | 0.33 |
| | Unpaved roads | 0.09 | Rainwater tank | 5% | 10 |
| **Green space** | Own vegetation | 1.12 | / | / | 0.95 |
| | Newly set green space | 0.34 | Sunken green space | 65% | |

permissible volume, thus reducing runoff. The stored rainwater can be utilized for watering plants and cleaning streets, which conserves water and enhances the living environment in older residential areas. Nonetheless, due to its limited capacity, this measure is often employed alongside other strategies.

Based on the functional classification of land in the community [27], LID facilities were set up in residential areas, office areas, outdoor activity areas, and green areas; the maximum land use ratios of LID facilities were determined in accordance with the abovementioned principles of LID layout and relevant technical specifications and literature. The parameters for each LID facility are presented in Table 2 and Table 3 [28–29].

## Optimization objectives and constraints

**1 Cost effectiveness.** The construction cost of a single LID measure is primarily composed of labor, material, machinery, and comprehensive expenses.

**Table 4. Cost List of LID Facilities.**

| LID facility | Green roof | Sunken green space | Rainwater tank | Permeable Pavement |
|---|---|---|---|---|
| Labor, Yuan/m² | 33.86 | 13.17 | / | 45.6 |
| Materials, Yuan/m² | 113.67 | 48.35 | 2000 | 210 |
| Machinery, Yuan/m² | 24.84 | 21.47 | / | 16.63 |
| Comprehensive cost, Yuan/m² | 49.27 | 27.79 | / | 70.9 |
| Total | 221.64 | 110.78 | 2000 | 343.13 |

$$\min(F_{total} = \sum_{i=1}^{n} C_i \cdot S_i), \tag{3}$$

$$C_i = a + b + c + d, \tag{4}$$

In the formulas,

$F_{total}$ is the total construction cost of LID facilities in the research area, in yuan;

n is the total number of LID facilities;

$C_i$ is the integrated cost value of different LID facilities, Yuan;

$S_i$ is the area covered by each LID measure, m²;

The costs associated with LID facilities, including labor, material, machinery, and comprehensive costs (Table 4), were determined based on typical community cases [30] outlined in the Cost Index Cases of Urban Public Facilities: Sponge City Construction Project, compiled by the Ministry of Housing and Urban–Rural Development.

**2. Cost effectiveness.**

(1) Average Total Runoff Reduction Rate

Overflow may occur in urban drainage networks when rainfall exceeds the bearing capacity of pipelines, while excessive overflow can lead to urban waterlogging. The rainwater storage capacity of LID facilities can be understood intuitively by comparing the changes in total runoff before and after sponge transformation.

$$\max\left(R_{total} = \frac{A}{i}\right), \tag{5}$$

$$A = \sum_{i=1}^{n} \frac{(R_{i\ original} - R_{i\ present})}{R_{i\ original}}, \tag{6}$$

In the formulas,

$R_{total}$ is the total runoff reduction rate, %;

A represents the runoff reduction rate of a single catchment area, %;

$R_{i\ original}$ denotes the runoff in the catchment area prior to the implementation of LID facilities, m3/s;

$R_{i\ present}$ indicates the runoff in the catchment area following the implementation of LID facilities, m3/s;

i refers to the number assigned to the catchment area.

(2) Average Node Overload Time

Node overload is defined as a situation in which the accumulation of water at a node surpasses the maximum permissible height at the top; overload time refers to the period during which the water level at a node exceeds the maximum height until it declines below this maximum threshold. Node overload easily causes water accumulation, resulting in surface runoff. Therefore, the fewer overloaded nodes there are and the shorter the overload time is, the better.

$$\min \left( T = \frac{\sum_{i=1}^{n} t_{overload}}{a} \right),$$  (7)

In the formula,
i is the number of overloaded nodes in the scheme;
a is the number of overloaded nodes without LID facilities;
T is the average overload time at a node, min;
i is the overload time at each node, min;

**3. Constraints.** Based on the maximum deployment ratio of LID facilities for each land use category, as outlined in Table 3, the area constraints are established as described in formulas 8 and 9.

$$0 < S_i < T_i \cdot L_i,$$  (8)

$$S_{rainwater\,tank} < 10,$$  (9)

In the formulas,
$S_i$ is the area (m$^2$) or quantity of each item of LID facilities;
$T_i$ is the area occupied by functional areas, m$^2$;
$L_i$ is the maximum proportion of LID settings, %.

To further evaluate the performance of each scheme along the Pareto curve in terms of environmental benefits, the actual purification capacity of LIDs in addressing air pollution was calculated using formula 10. Compared with rainwater tanks and permeable concrete, green roofs and sunken green spaces have been found to effectively purify air pollutants by reducing carbon dioxide levels, filtering ozone, and mitigating urban heat effects. Research has demonstrated that green roofs are more efficient at reducing air pollution than standard trees, although their cooling effect does not match that of trees [31]. According to the study conducted by Yu et al. [32], the median annual volume of air pollutants controlled by green roofs was chosen as a statistical measure. The control volumes for various pollutants are detailed in Table 5.

$$G = T * S,$$  (10)

In the formula,
G is the total amount of pollutants under control, t/km$^2$;
T is the value of each pollutant.

**Table 5. Pollutant Purification Amounts of Green Roofs.**

| Pollutant | Value, t/km$^2$ |
|---|---|
| NO$_2$ | 1.895 |
| O$_3$ | 3.66 |
| SO$_2$ | 1.55 |
| PM10 | 0.605 |

## Algorithm setting

The NSGA-II multi-objective genetic algorithm was utilized to model the evolutionary principles observed in natural populations [33]. The initial parameters were established as follows: population size, 100; total, 500 iterations; crossover probability, 0.9; and mutation probability, 0.1. When two individuals possessed the same level of dominance, priority was given to the exploration of regions with a higher crowding density. This approach aimed to maintain the diversity of the population and prevent convergence to local optima.

## Results

The coding work required in this study was completed in Python. With the help of the programming software, the calculation engine was directly called, and the calling process was integrated into the NSGA-II operation flow. The real-time interaction of the model simulation data was achieved by coupling the SWMM with the NSGA-II algorithm, thereby realizing automatic optimization of the objective functions. PySWMM is an SWMM interface package developed in Python that aims to establish, operate, and study complex urban pipeline systems. With PySWMM, users can browse and modify the SWMM, observe the changes in simulated values during simulation, and modify the relevant data of internal elements in the model. This study adopted the Python language and used the PySWMM package to call the SWMM model, integrating the SWMM simulation process into optimization. PyMOO is an open-source Python library that focuses on solving multi-objective optimization problems.It offers a set of flexible and efficient tools to assist researchers and developers in tackling complex issues, directly utilizing the NSGA-II algorithm and supporting custom ElementwiseProblem. Thereby constructing a true multi-objective optimization model for rainwater pipe networks.

Using Python programming software, the inp file from SWMM was accessed, and NSGA-II algorithms were applied for optimization debugging by PyMOO library. the NSGA-II algorithmrun took about 88 to 90 min on an AMD Ryzen9-7900X desktop personal computer with a 4.70 GHz processor,and the length of computational time depends on the performance of the computer processor. After completing 500 iterations, an optimal set of solutions was identified, considering LID construction costs, catchment area runoff reduction rates, and node overload times as the objectives for optimization. For each rainfall scenario, a Pareto optimal front was established, comprising 100 optimal LID distribution strategies based on dominance ranking (Fig 5). All strategies included in this curve were non-dominated. The red curve in Fig 6 represents the Pareto optimal frontfitted by simultaneously optimizing the three objective functions. The blue curve is the projection of the optimal curve on the xz plane, which is the curve of node overload time versus runoff reduction rate. The green curve displays the projection of the optimal curve in the xy direction, which is the node overload time–construction cost curve. The pink curve is the projection of the optimal curve in the yz direction, representing the construction cost–runoff reduction rate curve.

Fig 6 illustrates the cost-runoff curves fitted with recurrence periods of 3, 5, 10, and 20 years.

As indicated by the figure, there exists a direct relationship between the construction cost and the runoff control rate for each recurrence interval. However, the cost required to achieve an equivalent runoff control effect escalates with the lengthening of the recurrence period. For instance, for precipitation occurring every three years (P = 3), the construction cost for an LID scheme that achieves a 10% runoff reduction is only 979,500 yuan. When the recurrence period extends to five years (P = 5), the cost to reach this reduction rate increases to 1,154,900 yuan; for a ten-year recurrence period (P = 10), the construction of a similar LID scheme requires 1,459,600 yuan; and when the recurrence period reaches twenty years (P = 20), this cost increases to 1,810,300 yuan. Consequently, the construction cost increases at an accelerated rate with increasing precipitation intensity. Similarly, with a budget of 1.6 million yuan for LID construction in this area, a scheme designed for precipitation with a recurrence period of one year (P = 1) can achieve a runoff control rate of 14%; however, for a twenty-year recurrence period (P = 20), the same budget can only attain a runoff control rate of 9.6%, which signifies a 30% reduction from the control rate for the shorter recurrence period.

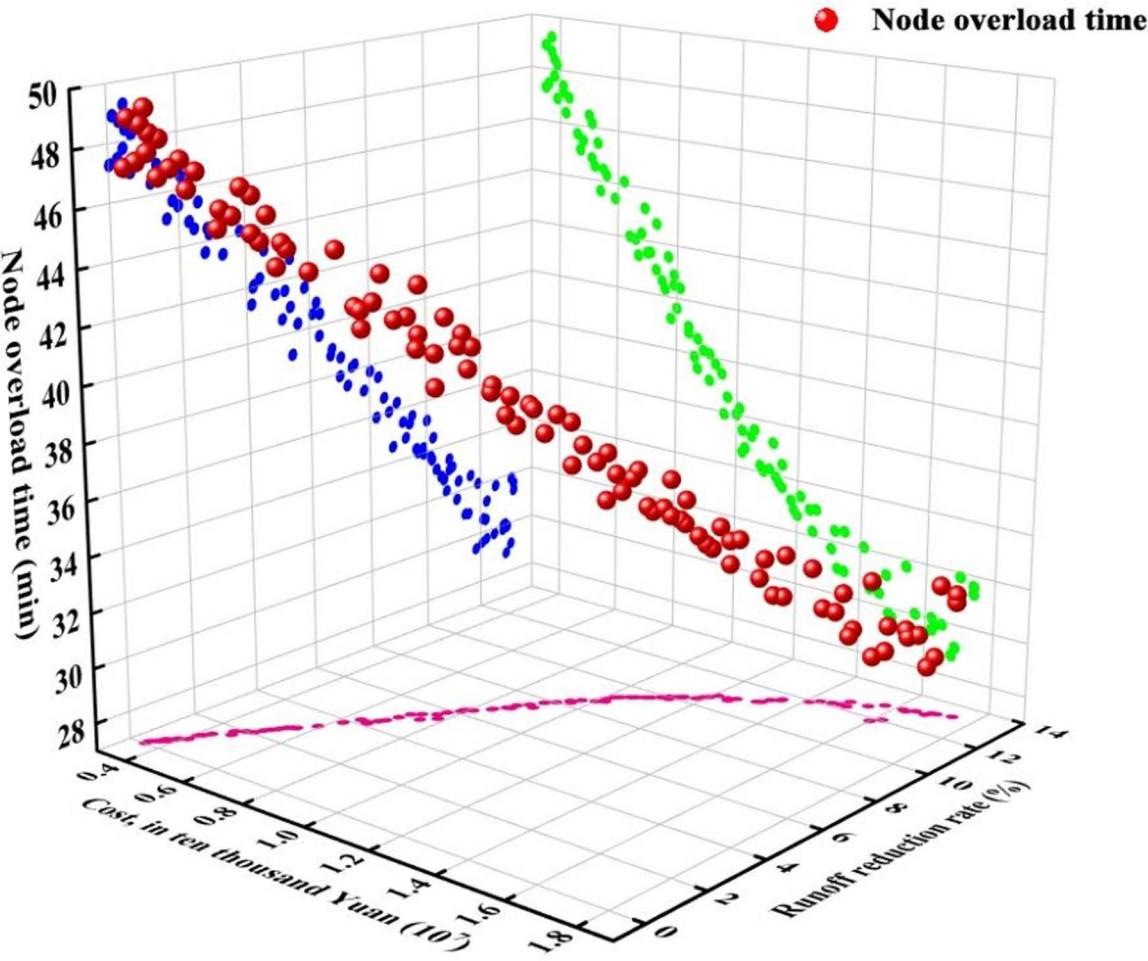

**Fig 5. Pareto optimal front when P is 5.**

Taking rainfall with a ten-year recurrence as an example, Fig 7 illustrates the Paretooptimal front determined by an algorithm. This curve presents an optimal balance of LID facilities offering the lowest construction costs, the highest runoff reduction rates, and the shortest node overload times for a precipitation event of 84.82 mm and a peak rainfall intensity of 0.4. Within this set of solutions, the scheme with the highest construction cost entails a cost of 1,950,000 yuan, a runoff reduction rate of 11.82%, and an average node overload time of 37.36 minutes. Conversely, the scheme with the lowest construction cost incurs a cost of 370,000 Yuan, achieving a minimal runoff reduction rate of 0.02% and an average node overload time virtually identical to that of scenarios lacking LID measures. The minimum average node overload time recorded is 36.21 minutes, corresponding to a construction cost of 1,770,000 yuan.

From the three-dimensional projections of the Pareto optimal front in the x-y-z directions, it is observed that a point situated closer to the bottom left corner signifies a solution with higher construction costs and a greater runoff reduction rate, whereas a point near the top right corner indicates a solution with lower costs and diminished runoff reduction effectiveness. After the runoff reduction rate surpasses 8%, the rate at which overall runoff decreases begins to slow. The node overload time changes in accordance with the variations in runoff volume. Within a 10% runoff reduction rate, the node overload time consistently diminishes as the reduction rate climbs. However, once the runoff reduction rate exceeds

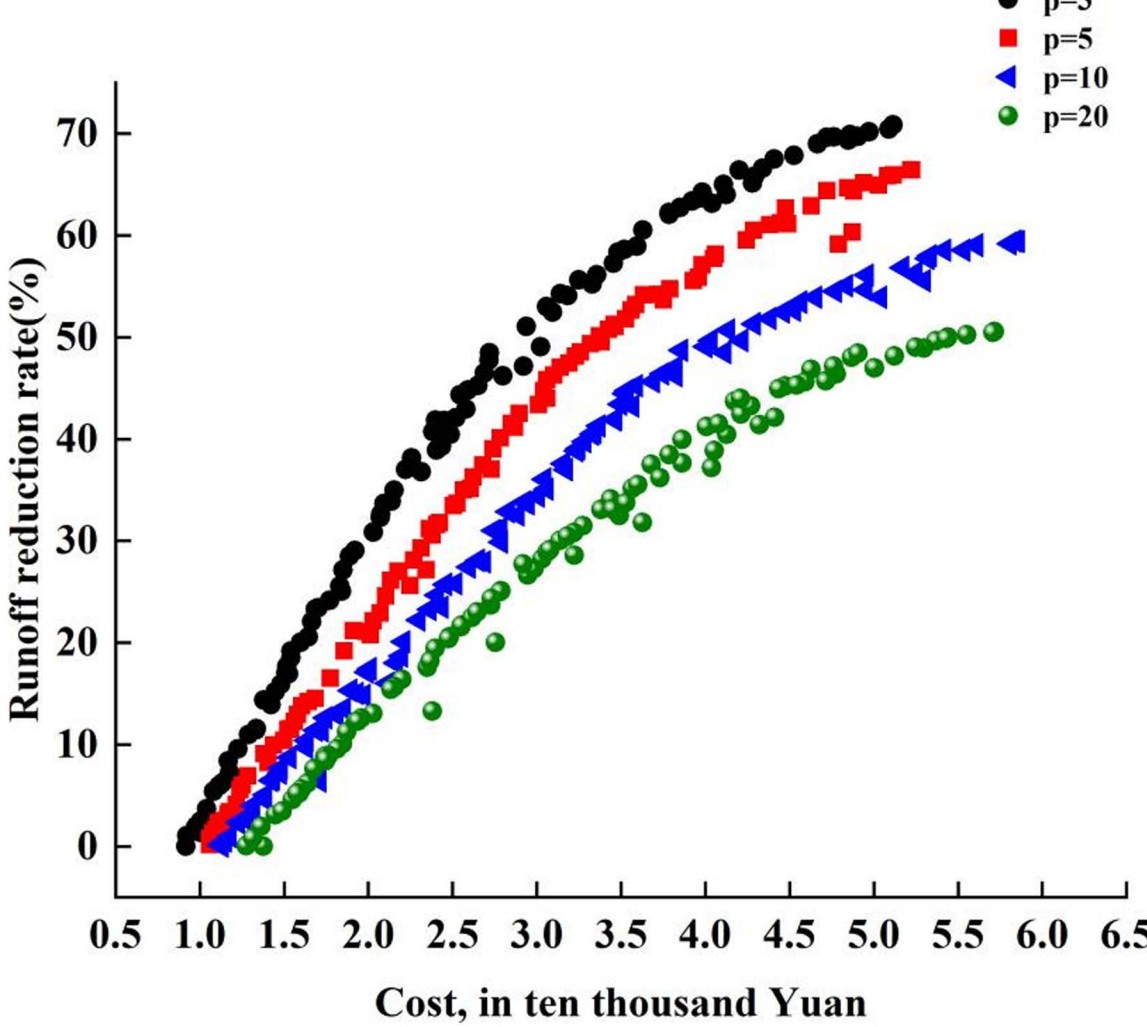

**Fig 6. Cost-runoff curves.**

10%, the duration for water to accumulate at a node, from reaching its peak to dropping below this maximum, does not significantly decrease and may actually show a slight increase. This phenomenon can occur because the incorporation of LID measures might lead to an increase in feature width and a decrease in the length of the central affluence channel, consequently shortening the time required for rainfall to reach the outlet of the area and increasing the likelihood of water accumulation. In essence, as the construction cost for LIDs increases, the node waterlogging time initially decreases but starts to increase after decreasing to a certain threshold.

For a precipitation event with a recurrence interval of ten years (P = 10),Five optimal solutions (Scheme I to Scheme V) for case were selected for further analysis which schemes were selected based on ascending construction costs. These schemes were then modeled using SWMM software to evaluate changes in catchment area performance, node response, and environmental benefits. The construction costs for these five LID schemes ranged from 700,000 Yuan to 1,770,000 Yuan, with runoff reduction rates varying between 6.48% and 11.82% and node overload times decreasing from 57.46 minutes to 55.91 minutes. Figs 8 and 9 display the volume of node waterlogging and the reduction in air pollutants with and without the implementation of LID facilities, respectively.

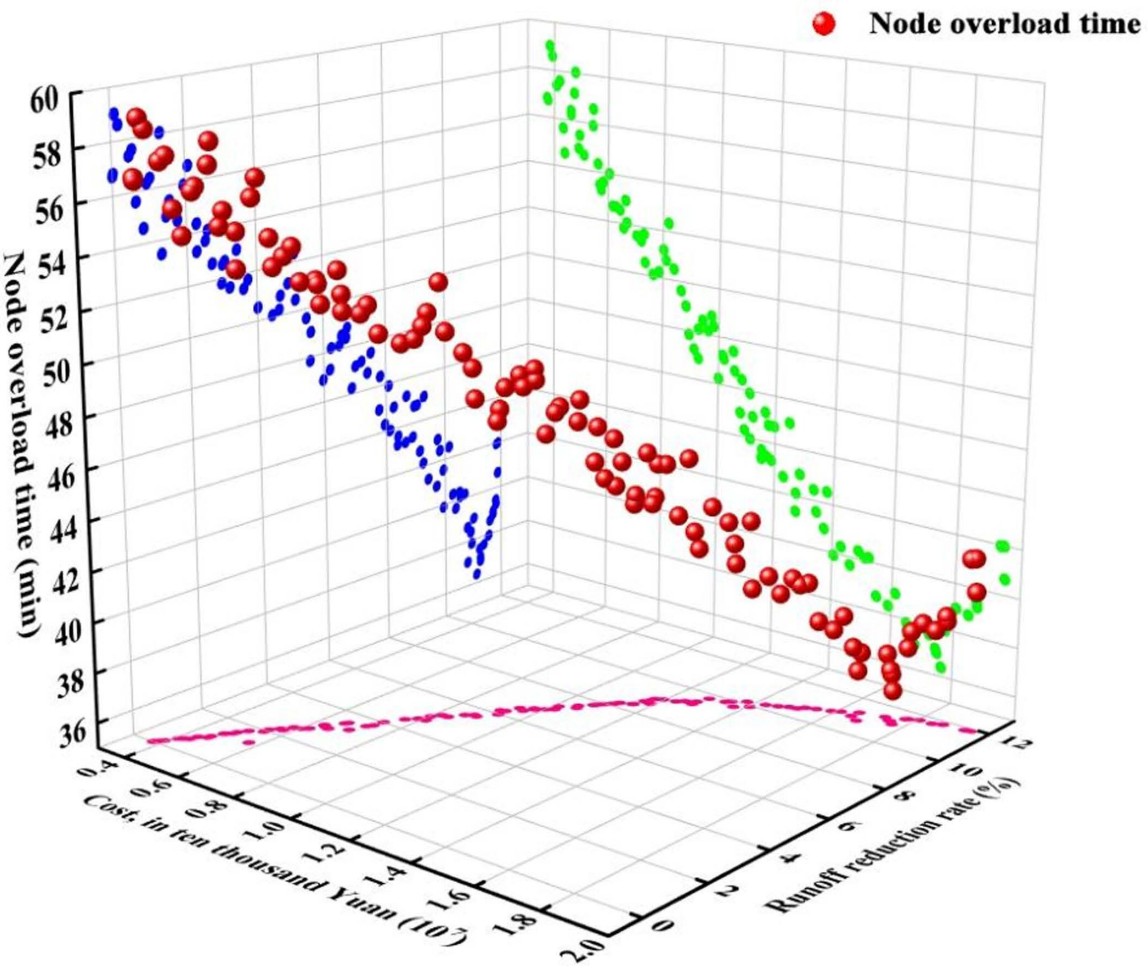

**Fig 7. Pareto optimal front when P is 10.**

As the construction costs for the five selected schemes progressively increase, a corresponding steady reduction in node waterlogging duration is observed, and the decline rate was relatively average. Among these schemes, Scheme V achieved the shortest waterlogging duration, with a reduction of 0.1 hours (6 minutes) compared to the control group, and exhibited the lowest number of waterlogging nodes, showing a decrease of 6 nodes relative to the control group. The inclusion of random mutation and recombination in the genetic algorithm results in variations in the coverage area of each LID facility, which, in turn, causes fluctuations in the number of waterlogged nodes. Thus, the effectiveness of a scheme is primarily judged based on the duration of waterlogging.

Moreover, the integration of LID infrastructures contributes to environmental relief by reducing the burden of harmful gases. The reduction in harmful gases achieved by the five schemes was determined using formula 10. This study specifically calculated the volume of air pollutants purified by green roofs, with the area of green roofs varying according to a predetermined probability. As illustrated in Fig 9, Schemes I, III, and IV demonstrate comparable effects on the reduction of harmful gases; Schemes II and V, which feature larger areas of green roofs, exhibit superior air purification performance. Extended waterlogging durations suggest inadequate rainwater drainage, increasing the likelihood of water accumulation points within the community, leading to traffic disruptions, increased vehicle emissions, and environmental harm. Conversely, a larger area covered by LID pavements can mitigate air pollution to a certain degree.

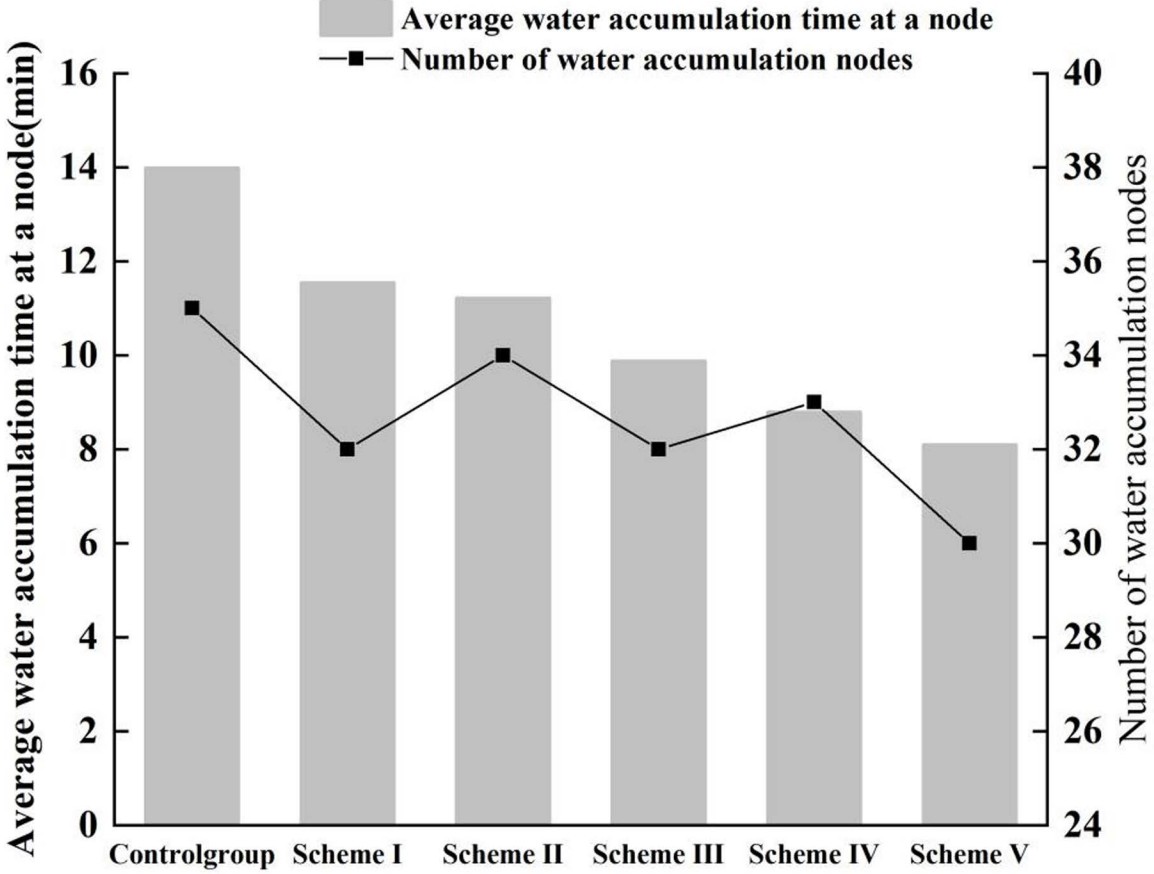

**Fig 8. Node waterlogging amount and quantity of waterlogging nodes.**

Given the performance of the five schemes in terms of waterlogging duration and air purification capabilities, Schemes II and V, with construction costs of 1,153,500 Yuan and 1,765,300 Yuan, respectively, were further analyzed for their impact on runoff curves in the catchment areas (Fig 10). For instance, in catchment area H00048, which includes residential building no. 2 and a vast sidewalk area, the algorithm opted to implement green roofs and pervious concrete to manage rainwater runoff. According to the runoff curve over a 120-minute rainfall event, runoff volumes under both schemes match those without LID measures for the initial 30 min. However, from the 40th minute onward, the runoff volumes under these schemes begin to diminish, peaking later. The peak flow rate in the control group for this catchment area was 4.74. Schemes II and V show reductions in peak runoff volumes of 10% and 15%, respectively, on an annual basis, along with a noticeable delay in peak occurrence time. In the control group, peak runoff occurred at the 52nd minute, whereas under Schemes II and V, peaks were reached at the 56th and 58th minutes, respectively.

Figs 11–13 demonstrate that, compared to that in the control group, surface runoff in the research area was significantly reduced through sponge transformation. The decrease in runoff is primarily attributed to absorption and storage by the vegetation present in LID facilities and, to a lesser extent, discharge into the rainwater drainage system through drainage pipes incorporated in pervious pavements. A minor fraction of the reduced runoff permeates into the ground, contributing to the groundwater reserves. There was no marked change in the infiltration volume, as the installation of LID facilities did not directly alter the infiltration behavior or rate of the soil layer.

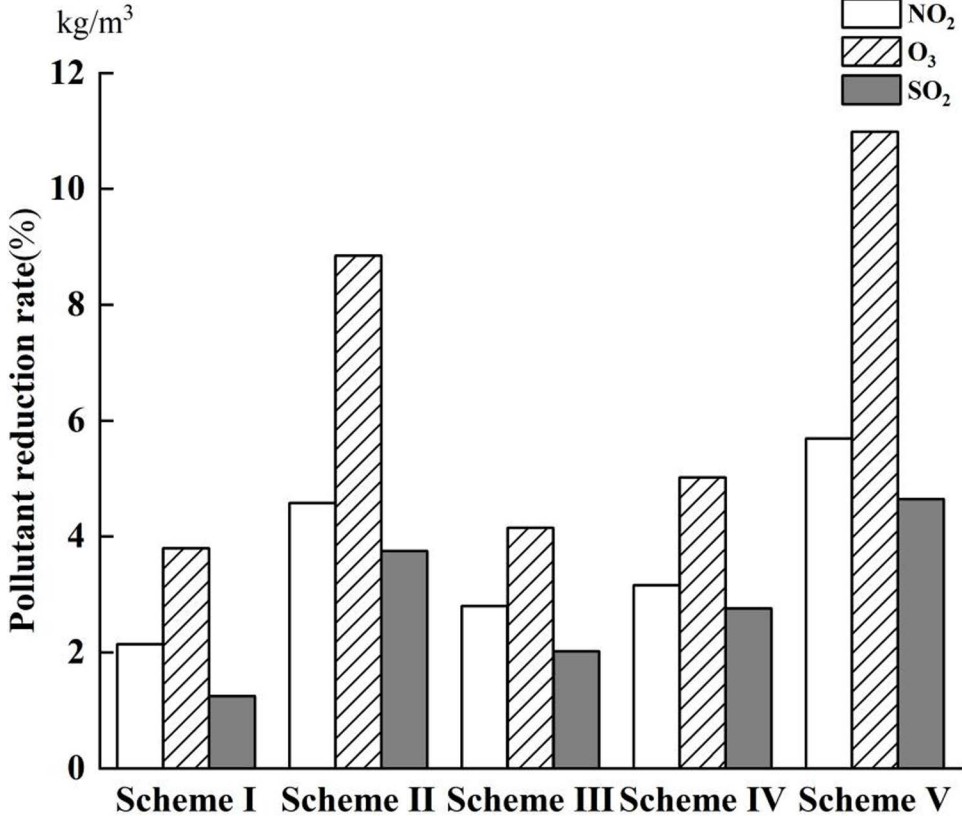

**Fig 9. Reduction of pollutants in the air.**

Optimal schemes with varying priorities can be identified by comparing the construction schemes across different cost ranges and analyzing detailed parameters of the catchment areas. When focusing solely on hydrological benefits, Scheme V stands out for its superior performance in terms of runoff reduction and the minimization of node waterlogging duration; however, its associated construction costs are significantly greater than those of other schemes. Therefore, when taking overall investment into consideration, Scheme II emerges as the optimal choice for the final transformation. Scheme II achieves similar levels of air purification and runoff control as the more expensive schemes. Under Scheme II, the area covered by a specific LID facility can be expanded as needed to achieve a targeted and sensible transformation of the research area.

## Conclusion and outlook

In this study, a stormwater model was developed using relevant rainfall data and detailed parameters of the research area. This model was then integrated with intelligent algorithms to identify optimal LID layout schemes across different recurrence periods, with a focus on the construction cost, runoff control rate, and node overload time as the primary objectives for optimization. Compared with traditional LID strategies, this integrated approach offers a more scientific and rational method of planning. The study culminated in the following conclusions:

1) Based on the collected data and drawings, the study area was divided into sub-areas using the Thiessen polygon method, and each hydrological cycle unit was established and defined to determine the hydrological connections

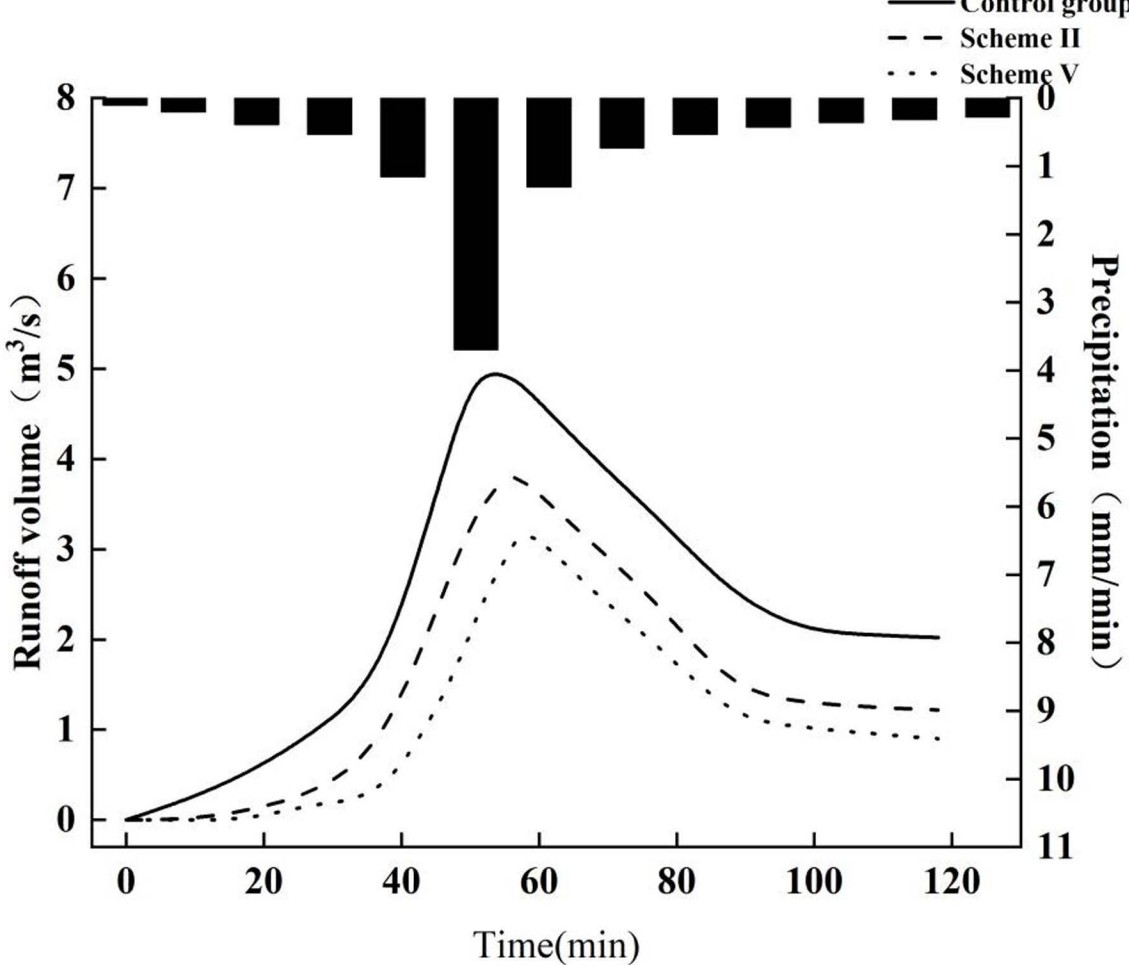

**Fig 10. Curve of precipitation and runoff volume.**

between the sub-catchment areas. The pipe network was generalized to include 58 rainwater pipes, 57 rainwater wells, and 2 discharge outlets.

2) To solve the waterlogging problem faced by the old residential community after rainstorms, this study adopted the Chicago rainfall type to simulate rainfall and selected P = 3, 5, 10, and 20 as the rainfall conditions. In this context, sponge city renovations were carried out for the residential community. Based on the characteristics of each LID and the needs of the old residential area, four renovation measures, i.e., green roofs, sunken green spaces, permeable pavements, and rainwater tanks, were selected for this study.

3) The implementation of green roofs, sunken green spaces, rainwater tanks, and pervious pavements can optimize drainage in catchment areas from various perspectives. Expanding pervious surfaces helps to reduce surface runoff in impervious areas, delay the onset of peak runoff, and diminish peak flow rates. Additionally, these facilities contribute to air quality purification, landscape enhancement, and the promotion of environmental consciousness within the community. Thus, sponge transformation is deemed essential for the rejuvenation of older residential areas.

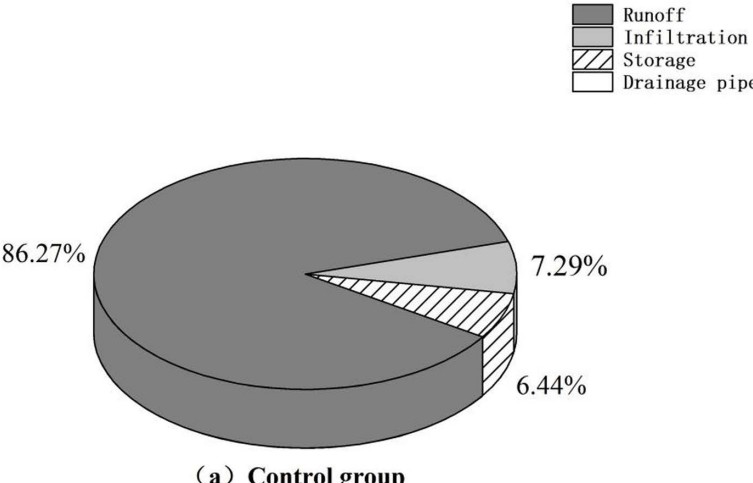

**Fig 11. Runoff analysis chart.** (a) Control group.

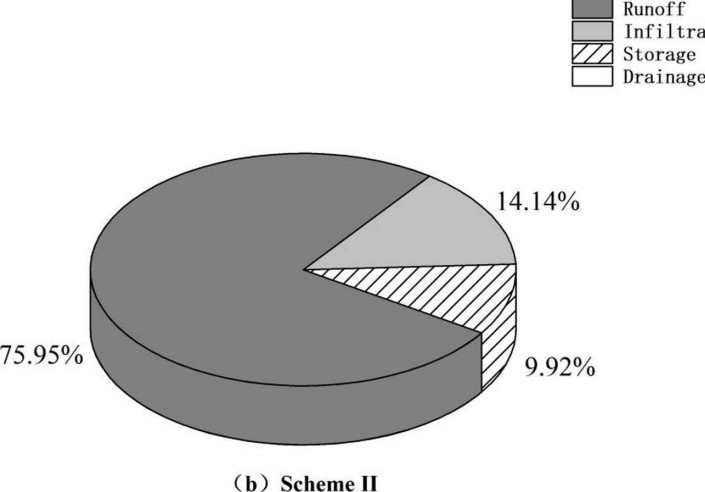

**Fig 12. Runoff analysis chart.** (b) Scheme II.

4) The effectiveness of stormwater control within a given recurrence interval generally improves with increasing construction costs. The average node overload time decreases with increasing construction cost and runoff reduction rate. The construction cost of LID reaches maximum efficiency when the controlled runoff quantity reaches 10% in a ten-year recurrence period(P = 10). However, the construction cost increases at a high speed when it exceeds 10%. This study is also applicable to cities or regions with strict economic constraints,because optimal schemes with varying priorities can be identified by comparing the construction schemes across different cost ranges and analyzing detailed parameters of the catchment areas. However, the final scheme should be selected based on a comprehensive evaluation of various factors, including land use type, climatic conditions, and policy decisions.

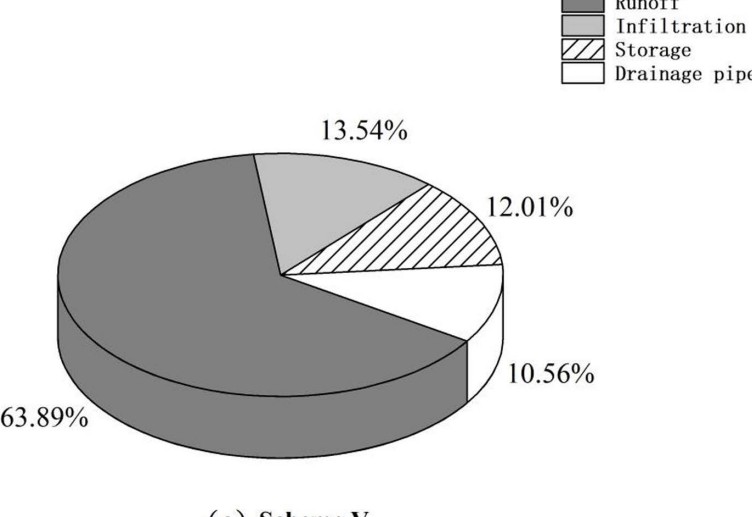

**Fig 13. Runoff analysis chart.** (c) Scheme V.

Given that the research area constitutes an old, compact urban residential community, the findings of this study are subject to certain limitations. Further research should be conducted in a larger administrative district,with experiments performed in different areas and comparisons made with results from other cities to validate the generalizability of the findings, and additional factors such as the urban water system layout and wastewater treatment and reuse need to be considered. The assumptions and parameter settings used in the simulations may deviate from reality, especially under extreme weather conditions. To address this limitation, we suggest that future research validate the reliability and accuracy of these simulation results under different environmental conditions through field experiments or long-term monitoring data. While simulations offer valuable insights for research, there are still more intricate factors that need to be taken into account in practical applications to enhance the model's applicability.

In order to control the long-term maintenance cost of LID, in the future design and construction stage, green roofs and sunken green spaces can be selected with vegetation that is adaptable to the local soil and climate, and has strong resistance to pests and diseases, to reduce the maintenance cost of vegetation in the later stage. For rainwater tanks and permeable paving, adopt drainage systems that are easy to clean and maintain, choose corrosion-resistant materials, and establish a detailed maintenance plan to regularly inspect and maintain the equipment and materials.Regarding the environmental benefits of LID, future studies should also consider the pollutant absorption capacity of vegetation within other LID facilities and the potential air pollution resulting from the materials employed in LID construction. For example, despite its beneficial properties, the installation of pervious concrete may generate dust and waste, in addition to its heat absorption capabilities and potential to interact with rainwater polymers, leading to precipitation.

## Author contributions

**Data curation:** Xinran Liu.

**Formal analysis:** Yijun Fan.

**Funding acquisition:** Yanjuan Li.

**Methodology:** Xinran Liu.

**Project administration:** Fuqin Liu.

**Resources:** Xinran Liu, Yanjuan Li.

**Software:** Yijun Fan.

**Supervision:** Fuqin Liu.

**Visualization:** Yijun Fan.

**Writing – original draft:** Yijun Fan.

**Writing – review & editing:** Yijun Fan.

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
