## [Decision Letter · Decision Letter 0]

9 Sep 2024

Dear Dr. Li,

Thank you for submitting your manuscript to PLOS ONE. After careful consideration, we feel that it has merit but does not fully meet PLOS ONE’s publication criteria as it currently stands. Therefore, we invite you to submit a revised version of the manuscript that addresses the points raised during the review process.

We look forward to receiving your revised manuscript.

Kind regards,

Namal Rathnayake, Ph.D.

Academic Editor

PLOS ONE

Journal Requirements:

2. Please note that PLOS ONE has specific guidelines on code sharing for submissions in which author-generated code underpins the findings in the manuscript. In these cases, all author-generated code must be made available without restrictions upon publication of the work. Please review our guidelines at https://journals.plos.org/plosone/s/materials-and-software-sharing#loc-sharing-code and ensure that your code is shared in a way that follows best practice and facilitates reproducibility and reuse

"This research was funded by National Natural Science Foundation of China (grant number 42101375) and The APC was funded by Fuqin Liu"

"I have read the journal's policy and the authors of this manuscript have the following competing interest"

5. We note that you have indicated that there are restrictions to data sharing for this study. PLOS only allows data to be available upon request if there are legal or ethical restrictions on sharing data publicly. For more information on unacceptable data access restrictions, please see http://journals.plos.org/plosone/s/data-availability#loc-unacceptable-data-access-restrictions. 

7. We note that Figure 1 in your submission contain map/satellite image which may be copyrighted. All PLOS content is published under the Creative Commons Attribution License (CC BY 4.0), which means that the manuscript, images, and Supporting Information files will be freely available online, and any third party is permitted to access, download, copy, distribute, and use these materials in any way, even commercially, with proper attribution. For these reasons, we cannot publish previously copyrighted maps or satellite images created using proprietary data, such as Google software (Google Maps, Street View, and Earth). For more information, see our copyright guidelines: http://journals.plos.org/plosone/s/licenses-and-copyright.

Additional Editor Comments:

The manuscript need to address the comments by the reviewers

Reviewers' comments:

Reviewer's Responses to Questions

**Comments to the Author**

1. Is the manuscript technically sound, and do the data support the conclusions?

Reviewer #1: Partly

Reviewer #2: Yes

2. Has the statistical analysis been performed appropriately and rigorously?

Reviewer #1: N/A

Reviewer #2: Yes

3. Have the authors made all data underlying the findings in their manuscript fully available?

Reviewer #1: No

Reviewer #2: Yes

4. Is the manuscript presented in an intelligible fashion and written in standard English?

Reviewer #1: No

Reviewer #2: Yes

Reviewer #1: Interesting topic. However, there are many slips in this paper. Please see my comments.

1. Abstract has to be formatted as per the journal.

2. There are several issues with citations throughout the manuscript.

"Furthermore, the proliferation of impermeable surfaces reduces rainwater infiltration, leading to lower groundwater levels and increased surface runoff2"

"Randall et al. 9 found through simulation that"

Please use the correct style.

3. It is very much interesting to see how you linked SWMM to NSGA II in the events of no library files. Can you briefly explain this.

4. What is the computational time for the 100 iterations?

5. It would be better to identify few solutions from the Pareto curve and then to discuss the real behaviour of them in depth!

6. Please see the following paper too...https://doi.org/10.3390/hydrology2040230

Reviewer #2: The study focuses on a specific historical urban area and relies on the characteristics of that region, making it unclear whether the findings can be applied to other areas. In particular, there is room for debate as to whether the same results can be obtained in regions with different climate conditions and terrain. Unless experiments are conducted over a broader range and comparisons are made with other cities, the generalizability of the results may be lacking. This issue should be addressed in the manuscript.

Although the potential for construction costs to become very high is indicated, it is unclear whether the balance between cost and effect has been sufficiently discussed. In particular, there seems to be a lack of consideration regarding the applicability in cities or regions with strict economic constraints, as well as the long-term maintenance costs. Please expand on this point.

The data and simulation results used are based on specific weather conditions and models, but it is uncertain whether they can be generalized to all conditions. In particular, results that depend on simulations may not necessarily match real-world applications. Please provide additional details addressing these concerns.

**Do you want your identity to be public for this peer review?** For information about this choice, including consent withdrawal, please see our Privacy Policy

Reviewer #1: No

Reviewer #2: No

---

## [Author Response · Author response to Decision Letter 0]

28 Oct 2024

Please see the attachement for detailed response.

---

## [Decision Letter · Decision Letter 1]

30 Dec 2024

Research on low-impact development layouts in old urban residential communities based on multi-objective optimization

PONE-D-24-25100R1

Dear Dr. Li,

We’re pleased to inform you that your manuscript has been judged scientifically suitable for publication and will be formally accepted for publication once it meets all outstanding technical requirements.

Kind regards,

Namal Rathnayake, Ph.D.

Academic Editor

PLOS ONE

Additional Editor Comments (optional):

Reviewers' comments:

Reviewer's Responses to Questions

**Comments to the Author**

Reviewer #1: (No Response)

Reviewer #2: All comments have been addressed

2. Is the manuscript technically sound, and do the data support the conclusions?

Reviewer #1: (No Response)

Reviewer #2: Yes

3. Has the statistical analysis been performed appropriately and rigorously?

Reviewer #1: (No Response)

Reviewer #2: Yes

4. Have the authors made all data underlying the findings in their manuscript fully available?

Reviewer #1: (No Response)

Reviewer #2: Yes

5. Is the manuscript presented in an intelligible fashion and written in standard English?

Reviewer #1: (No Response)

Reviewer #2: Yes

Reviewer #1: Authors have addressed the comments successfully, therefore, I would like to accept the revisions. Therefore, the editors can take a decision.

Reviewer #2: The paper is fully qualified for publication, with well-argued and clear explanations of the reviewers' points.

**Do you want your identity to be public for this peer review?** For information about this choice, including consent withdrawal, please see our Privacy Policy

Reviewer #1: No

Reviewer #2: No

---

## [Editor Report · Acceptance letter]

PONE-D-24-25100R1

PLOS ONE

Dear Dr. Li,

I'm pleased to inform you that your manuscript has been deemed suitable for publication in PLOS ONE. Congratulations! Your manuscript is now being handed over to our production team.

Kind regards,

on behalf of

Dr. Namal Rathnayake

Academic Editor

PLOS ONE